# Antiproliferative Activity of Mycelium vs. Fruiting Body: *Ganoderma subincrustatum* and *G. weberianum* from Sonora, Mexico

Damian López-Peña [1,2], Heriberto Torres-Moreno [3], Max Vidal-Gutiérrez [4], Ramón Enrique Robles-Zepeda [5], Aldo Gutiérrez [1] and Martín Esqueda [1,*]

[1] Research Center for Food and Development, A.C. Carr. Gustavo Enrique Astiazarán Rosas 46, La Victoria, Hermosillo 83304, Mexico; damian.lopez@uacj.mx (D.L.-P.); asaldana@ciad.mx (A.G.)

[2] Institute of Biomedical Sciences, Autonomous University of Ciudad Juárez, Av. Benjamín Franklin 4650, Condominio La Plata, Ciudad Juárez 32310, Mexico

[3] Northern Regional Unit, University of Sonora, Av. Universidad e Irigoyen, Caborca 83600, Mexico; heriberto.torres@unison.mx

[4] Southern Regional Unit, University of Sonora, Blvd. Lázaro Cárdenas del Río 100, Francisco Villa, Navojoa 85880, Mexico; max.vidal@unison.mx

[5] Central Regional Unit, University of Sonora, Blvd. Luis Donaldo Colosio y Rosales s/n, Centro, Hermosillo 83000, Mexico; robles.zepeda@unison.mx

[*] Correspondence: esqueda@ciad.mx; Tel.: +52-(662)289-2400 (ext. 504); Fax: +52-(662)280-0422

**Abstract:** The genus *Ganoderma* has been little studied in arid areas worldwide. *Ganoderma subincrustatum* and *Ganoderma weberianum* strains were obtained from the Sonoran Desert, Sonora, Mexico. *Ganoderma* spp. synthesize triterpenoids such as ganoderic acids with antiproliferative activity because they inhibit specific targets, induce apoptosis, and increase the activity of killer cells. Mycelium and fruiting body chloroform extracts from *G. subincrustatum* and *G. weberianum* were tested on HeLa, A549, L-929, and RAW 264.7 cell lines. Extracts from the fruiting body present higher antiproliferative activity than mycelium. All extracts induced vesicle and cellular debris formation in all cell lines, being non-selective for cancerous cells. Chloroform extract from *G. subincrustatum* fruiting bodies presented higher activity against all cell lines. Fractions F7 and F15 from this extract exhibited an $IC_{50}$ of 37.9 and 41.9 μg/mL on the A549 cell line, respectively; however, chloroform crude extract showed higher activity ($IC_{50}$ of <25 μg/mL) in all cell lines. Flow cytometry assays of F7 revealed cell death by apoptosis in A549 cells. NMR suggested the presence of ganoderic acids in F7. In future research, it will be interesting to characterize these fractions (metabolites, their bioactivities, and mechanism of action).

**Keywords:** *Ganoderma* triterpenes; anticancer drugs; fungal extracts; cancer cell lines

## 1. Introduction

Some species of *Ganoderma*, mainly *Ganoderma lucidum*, have been used in traditional Asian medicine for more than six millennia to modulate immunological activity and to treat hypertension, arthritis, asthma, anorexia, hepatitis, cancer, and other illnesses [1–3]. The study of natural products is the basis for the discovery of new molecules with different structures and bioactivities. It allows improving the drug design for effective anticancer therapies [4,5]. Cancer is the most prevalent disease worldwide; according to the World Health Organization, it is the first or second highest cause of death before the age of 70 [6]. Since cancer is characterized by uncontrolled cell proliferation, angiogenesis, and apoptosis evasion, the discovery of novel molecules and extracts to induce programmed cell death is relevant. Thus, apoptosis is one of the major targets for non-surgical cancer treatment [7].

*Ganoderma* has a cosmopolitan distribution, comprising more than 450 species by the year 2022, many of them unstudied (www.indexfungorum.org/names/Names.asp;

accessed on 04/September/2023). The most studied species is *G. lucidum*; however, metabolites and bioactivities have been reported for another 24 species. Some of the studied species are *G. amboinense*, *G. applanatum*, *G. boninense*, *G. capense*, *G. lipsiensis*, *G. neojaponicum*, *G. orbiforme*, *G. pfeifferi*, *G. sinense*, *G. tropicum*, *G. tsugae*, among others [8,9]. Isolated compounds from *G. weberianum* have antidiabetic and antiproliferative activity [10]. This indicates the lack of information on more than 400 species of *Ganoderma* genus.

*Ganoderma* species synthesize biologically active compounds, mainly lanostane-type triterpenoids (e.g., ganoderic acids: GA; triterpene alcohols), which exhibit antiproliferative activity against different cell lines [11–13]. *Ganoderma* triterpenoids are synthesized via the mevalonate/isoprenoid pathway from acetyl-coenzyme A [14]. Stages of conversion, construction, condensation, and post-modification are involved in its biosynthesis. The final stage, once lanosterol is formed, includes acylation, oxidation, and reduction reactions, to produce a complex group of molecules with different functional groups, like hydroxyl, carboxyl, and acetoxy [2,8]. Due to the relevance of *Ganoderma* triterpenoids, genes involved in this pathway have been studied [15].

For *G. lucidum* the expression of genes (*HMGR*, *FPS*, *SQS*, *LS*) related to the mevalonate/isoprenoid pathway increases during primordia development, which is correlated with the highest triterpenoid content in this phase of its life cycle [16]. Likewise, it has been reported that secondary mycelium contains mainly 3α-substituted compounds while the fruiting body presents mainly 3keto- or 3β-substituted compounds [17], which could be related to its bioactivity. Traditionally, *Ganoderma* cultivation used wood logs outdoors. Several decades ago, a modern indoor cultivation technique was developed using wood sawdust and agricultural by-products [18]. Because fruiting body production is a time-consuming activity for obtaining products from *Ganoderma* (months), the production of mycelia by liquid culture seems to be a promising technique; because it offers inter-batch homogeneity, less time is required, and fermentation factors can be directed toward the production of bioactive compounds [19,20].

*Ganoderma weberianum* is a poorly studied species with few reports, while for *G. subincrustatum* we did not find any work on the profile of its bioactive compounds with antiproliferative activity. Both species are distributed in Sonora, Mexico; therefore, to take advantage of our natural resources, this work aims to generate knowledge about the antiproliferative activity of chloroform extracts from the mycelium and fruiting bodies of *G. subincrustatum* and *G. weberianum* against different cell lines. Furthermore, apoptosis induction and nuclear magnetic resonance (NMR) of the bioactive fraction were addressed.

## 2. Materials and Methods

### 2.1. Fungal Strains

*Ganoderma subincrustatum* was collected from a live *Prunus persica* tree in a peach orchard at La Costa de Hermosillo, while *G. weberianum* was taken from a *Quercus* sp. tree in La Sierra de Mazatán, Ures, Sonora, México. Both species were previously recorded by López-Peña et al. [21,22]. Strains were deposited in the Fungus Collection of the Plant and Fungi Biotechnology Laboratory of the Research Center in Food and Development A.C.; *G. subincrustatum* was registered as BH-1 and *G. weberianum* as BH-21. *Ganoderma* strains were maintained on malt extract agar in Petri dishes.

### 2.2. Liquid and Solid Culture

Liquid medium composition for mycelium production comprised the following: glucose (16 g/L), peptone (2.93 g/L), corn flour (21 g/L), and soybean protein powder (7 g/L) [23]. A total of 400 mL of culture medium was inoculated with 16 agar disks 1 cm in diameter with mycelium for each strain. Liquid fermentation was maintained at $28 \pm 1$ °C, shaking at 120 rpm, initial pH 5.5, for 13 d in the dark. Mycelium was recovered by centrifugation and washed with distilled water. Fruiting body production was carried out using oak wood chips. The sterilized substrate was inoculated with spawn (wheat grain at 5% *w/w*) previously prepared. Incubation was conducted at $27 \pm 1$ °C in the

dark. Fruiting body development conditions were maintained at $27 \pm 1\ °C$, 90% relative humidity, light intensity of 48–58 $\mu mol \cdot m^{-2} \cdot s^{-1}$, and 500 to 1000 ppm of $CO_2$ in air [24]. The fruiting body was harvested when the pileus fully developed or stopped growing. Mycelia and fruiting bodies were freeze-dried and stored in a dry place until use.

### 2.3. Extracts

Mycelia and dried fruiting bodies were ground with an electric mixer to obtain a powder, from which 1 g of sample was extracted with 95% ethanol (30 mL) by sonication for 45 min (Ultra Sonic Cleaner, Branson 2210R-MT), with a power of 210.6 W and operating frequency of 40 kHz. Biomass was removed by centrifugation and filtration; then, the extracts were concentrated under reduced pressure conditions (BÜCHI Rotavapor, RE 121). Afterwards, dried residues were suspended in distilled water and then extracted with chloroform; the chloroform layer was recovered [23], then dried at 30 °C and stored at −20 °C until bioactivity assays. The extracts obtained were the following: Gsm (*G. subincrustatum* mycelial extract), Gsfb (*G. subincrustatum* fruiting body extract), Gwm (*G. weberianum* mycelial extract), and Gwfb (*G. weberianum* fruiting body extract).

### 2.4. Bioguided Fractionation and Chemical Analysis

*Ganoderma subincrustatum* fruiting body extract (2.2 g) was fractioned by column chromatography (5.5 × 50 cm) using Sigma Aldrich silica gel 60 (200–400 mesh). The column was eluted with ethyl acetate-methanol (EtOAc-MeOH) in different proportions (*v/v*, 100:0, 95:5, 90:10, 85:15, 80:20, 50:50, 0:100). Three hundred thirty-three fractions (10 mL) were collected and analyzed by thin-layer chromatography (TLC). The spots on the TLC were visualized under UV light (254/366 nm) then stained with *p*-anisaldehyde-$H_2SO_4$-EtOH (1:1:98), followed by heating at 110 °C until positive areas appeared [25]. Twenty-six (F1 to F26) subfractions were grouped according to TLC patterns: F1 (8.2 mg), F2 (83.1 mg), F3 (13.4 mg), F4 (27.8 mg), F5 (323.8 mg), F6 (93.3 mg), F7 (114.2 mg), F8 (6.2 mg), F9 (37 mg), F10 (12.7 mg), F11 (34.3 mg), F12 (22.6 mg), F13 (61.7 mg), F14 (8.5 mg), F15 (39.2 mg), F16 (11.8 mg), F17 (44.3 mg), F18 (18.7 mg), F19 (14.3 mg), F20 (7.8 mg), F21 (9.3 mg), F22 (17.4 mg), F23 (18.2 mg), F24 (30.8 mg), F25 (55 mg), and F26 (337. 8 mg).

The chemical analysis of the F7 subfraction was performed by $^1H$ and $^{13}C$ NMR analysis (nuclear magnetic resonance). The NMR spectra were recorded at 400 MHz for $^1H$ and 100.6 MHz for $^{13}C$ in an Agilent 400 MHz NMR Magnet Fourier transform instrument. The spectra were recorded in $CD_3Cl$ (Chloromethane-$D_3$) at 25 °C and the chemical shift was expressed in δ values (ppm); TMS (tetramethylsilane) was used as an internal reference [26].

### 2.5. Cell Culture

Cell lines were cultured in DMEM (Dulbecco's Modified Eagle Medium), supplemented with 5% heat-inactivated fetal bovine serum and penicillin-streptomycin (100 U/mL) in 25 $cm^2$ culture dishes, which were kept in an incubator isotherm (Fisher Scientific, Waltham, MA, USA) with 5% $CO_2$, at 37 °C and 95% relative humidity. Cancer cell lines HeLa (human cervical carcinoma), A549 (human lung carcinoma), and non-cancerous cell line L-929 (subcutaneous connective tissue) were purchased from the American Type Culture Collection (ATCC, Gaithersburg, MD, USA). The RAW 264.7 (macrophages transformed by the virus *Albeson leukemia*) line was kindly provided by Dr. Emil R. Unanue (Department of Pathology and Immunology, Washington University in St. Louis, MO, USA).

### 2.6. Antiproliferative Activity

The antiproliferative activity of the chloroform extracts and chromatographic fractions was determined using the MTT (3-(4,5-dimethylthiazol-2-yl)-2,5-diphenyltetrazolium bromide) assay (Cell proliferation Kit I, Roche), according to Müller et al. [27], with some modifications. The MTT assay is based on the ability of viable cells to reduce MTT to formazan through mitochondrial dehydrogenases. Briefly, cell suspensions of 200,000 cells/mL were

placed in 96 ELISA plates (Costar, Corning, NY, USA) and incubated for 24 h. Afterward, extracts were dissolved in dimethyl sulfoxide (DMSO) and redissolved in DMEM to give a final DMSO concentration of 0.25%, showing no interference with the assay. Extract evaluations (25–100 µg/mL) were performed over 48 h. After the treatment, morphological changes were observed with an inverted microscope and microphotographs were taken with the help of a digital camera. Finally, according to the manufacturer's instructions, 10 µL of MTT solution was added to each well and incubated for 4 h in a humid atmosphere at 37 °C. A total of 100 µL of the solubilization buffer was added to the wells and incubated overnight, and absorbance measurements were performed at 570 and 655 nm in a microplate reader. The antiproliferative activity was determined as $IC_{50}$ (concentration of extract or fraction that inhibits the 50% of cell proliferation) value using GraphPad Prism 5 (GraphPad Software, Inc., San Diego, CA, USA).

### 2.7. Flow Cytometry

The apoptotic effect of subfraction F7 on A549 cells was determined by double staining with annexin V-FITC (AN) and propidium iodide (PI) (Sigma-Aldrich, Saint Louis, MO, USA). Briefly, $2 \times 10^5$ cells/mL were placed in a 12-well cell culture plate (Costar, Corning, NY, USA) and incubated for 24 h. Cells were then treated with F7 (40 µg/mL) for 24 h. The antineoplastic doxorubicin hydrochloride (DOXO) (1 µM) was used as a positive control. After treatment, cells were collected and washed twice with PBS and then resuspended for 10 min in a staining solution containing AN (1 µg/mL) and PI (10 µg/mL). Finally, $1 \times 10^5$ cells were acquired by fluorescence-activated cell sorting (FACS) (Canto II FACS, Becton Dickinson, CA, USA) according to Torres-Moreno et al. [26]. A dot plot was introduced to identify viable cells, dead cells, cells in early apoptosis, and cells in late apoptosis. Viable cells were recognized as doubly negative for AN and PI (AN−/PI−), dead cells were negative for AN and positive for PI (AN−/PI+), cells in early apoptosis were identified as AN positive, and PI negative (AN+/PI−), whereas cells in late apoptosis were recognized as being doubly positive for AN and PI (AN+/PI+). The cells in each quadrant were expressed as a percentage of the total number of stained cells.

### 2.8. Statistical Analysis

A one-way analysis of variance was applied to the antiproliferative activity and apoptosis induction. The Tukey-Kramer test was performed to compare treatments with their respective controls ($p \leq 0.05$). Data are presented as mean ± standard deviation.

## 3. Results and Discussions

### 3.1. Antiproliferative Activity of G. subincrustatum and G. weberianum

Fruiting body extracts showed higher bioactivity than mycelium extracts in all evaluated cell lines, although we used a complex culture media with acceptable GA production in *G. lucidum* [23]. Gsfb extract exhibited higher activity than Gwfb against all cell lines. The lower activity was shown in Gwm with an $IC_{50} \geq 100$ (µg/mL) in all cell lines. The non-cancerous cell line L-929 was highly affected by Gsfb and Gwfb, which suggests that the antiproliferative activity of Gsfb and Gwfb extracts is non-selective for cancer cell lines (Table 1). This could be attributed to a higher production of bioactive compounds in fruiting bodies, as was also observed by Chen et al. [16].

Crude extracts with $IC_{50}$ of ≤30 µg/mL are considered promising, according to the National Cancer Institute [28]. Several extracts, fractions, and isolated compounds from *Ganoderma* spp. have proved to exert antiproliferative activity against different cell lines. Zolj et al. [29] tested a commercial product from *G. lucidum* spores (ReishiMax GLp) with 6% triterpenes content. They obtained an antiproliferative activity at 0.4–10 mg/mL (24–120 h treatments) on the human non-small cell lung adenocarcinoma (NCI-H1793) cell line. Liu et al. [30] investigated the antiproliferative activity of ethanolic extracts from *G. lucidum* and *G. sinensis* against different tumor cell lines and obtained higher $IC_{50}$ values than the present study (71–378 µg/mL). Nevertheless, the extracts tested in the present study were

designed to obtain a triterpenoid-enrich fraction [23], as we partitioned the crude ethanolic extract between water and chloroform, and the chloroform layer was recovered for the first stage of antiproliferative assays.

**Table 1.** Antiproliferative activity of *Ganoderma* spp. strains against several cell lines: mycelium vs. fruiting body.

| Extracts | Cell Lines IC$_{50}$ (µg/mL) | | | |
|---|---|---|---|---|
| | **HeLa** | **RAW 264.7** | **A549** | **L929** |
| Gsm | ND | 81.9 ± 4.3 | ND | ND |
| Gsfb | <25 | <25 | <25 | <25 |
| Gwm | ND | ND | ND | ND |
| Gwfb | 57.7 ± 6.8 | <25 | 42.8 ± 5.5 | <25 |

IC$_{50}$ is represented as mean ± standard deviation. Chloroform extracts: Gsm (*G. subincrustatum* mycelium), Gsfb (*G. subincrustatum* fruiting body), Gwm (*G. weberianum* mycelium), and Gwfb (*G. weberianum* fruiting body). ND: not determined to 100 µg/mL.

After 48 h of treatment, Gsfb and Gwfb induced vesicle and debris formation in all cell lines, being more evident in the A549 cell line (Figure 1). Apoptosis is the main target for anticancer therapy and some of the morphological hallmarks of apoptosis induction are cell shrinkage, fragmentation, formation of apoptotic bodies, and phagocytosis by adjacent cells [31]. Crude extracts and different triterpenoids isolated from *Ganoderma* spp. induce morphological changes related to apoptosis on several cell lines. Jang et al. [32] evaluated an ethanol–water (25/75) extract of *G. lucidum* on human gastric carcinoma cells. They observed morphological changes with treatment (1.5% for 72 h), like cytoplasmatic filaments and apoptotic bodies. Methanolic extract (80%) of *G. applanatun* also induced morphological changes in a human colon cancer cell line (Caco-2), at 80 and 160 µg/mL [33]. Furthermore, ganoderic acids Mf and S induce morphological changes and apoptosis in HeLa cells [34]. These results were the basis for choosing Gsfb extract for further analysis, including column chromatography to obtain different fractions and antiproliferative activity of those fractions on A549 cell lines.

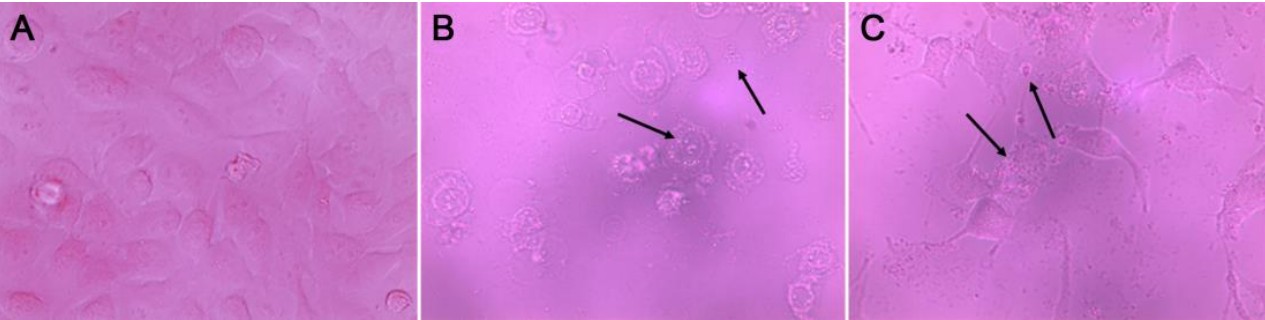

**Figure 1.** A549 cell line morphology. (**A**) Control, (**B**) *Ganoderma weberianum* fruiting body extract (100 µg/mL), (**C**) *Ganoderma subincrustatum* fruiting body extract (100 µg/mL). 40× magnification. Arrows indicate vesicle and debris formation.

### 3.2. Bioactive Fractions of G. subincrustatum

A column chromatography assay yielded 333 fractions (10 mL), which were grouped into 26 fractions, according to TLC patterns. Then, the antiproliferative activity of the 26 chromatographic grouped fractions (F1–F26) was tested against the A549 cell line. F7 and F15 showed significant differences at the evaluated concentrations (50–6.25 µg/mL), compared to the control ($p < 0.05$) with IC$_{50}$ values of 37.9 ± 6.2 and 41.9 ± 4.9 µg/mL, respectively (Figure 2). The IC$_{50}$ values of both fractions were higher than those of the chloroform extract. This was probably caused by the separation of some bioactive molecules by column fractionation. Due to the significance of lung cancer worldwide, the A549 cell line

has been used to assess the antiproliferative activity of several products through numerous studies. The leaf extracts of *Sarcocephalus pobeguinii* showed a correlation between free radical scavenging capacity and the antiproliferative activity against A549 cells ($R^2 = 0.99$), with an $IC_{50}$ of 50.46–857.25 µg/mL [28], while in another study the extracts of *Populus niga* buds showed an $IC_{50}$ of 72.49 µg/mL [35]. Novel triazole hybrids of myrrhanone C, a natural polypodane bicyclic triterpene, the oxime-based triazole 4a showed potent activity vs. the A549 cell line with an $IC_{50}$ of 6.16 µm. This compound arrested the cell cycle in the G2/M phase and induced apoptosis [36]. Novel 1,2,3-triazole hybrids of myrrhanone B, compounds 30 and 29 showed activity vs. the A549 cell line with $IC_{50}$ values of 15.34 ± 0.37 and 28.12 ± 2.64 µm, respectively; compound 29 increased to 40.8% the SubG1 cells at 10 µm, which induced cell apoptosis [37]. Furthermore, titanium dioxide nanoparticles ($TiO_2$ NPs) (5 nm) induced cytotoxicity, DNA damage, and apoptosis in the A549 cell line in a concentration range of 50 to 200 µg/mL for 48 h. Flow cytometric analysis showed a $G_2$/M phase arrest and a higher percentage of apoptotic cells [38]. Carvacrol nanoemulsion induced apoptosis through increasing apoptotic proteins (Bax, caspase 3, caspase 9, cytochrome C), cell cycle arrest (reducing CDK2, CDK4, cyclin E, cyclin D1), and inhibiting autophagy (reducing the conversion of LC-3) into the doxorubicin-resistant A549 cell line [39]. Because of the aforementioned, we selected F7 for apoptosis analyses by flow cytometry and NMR characterization.

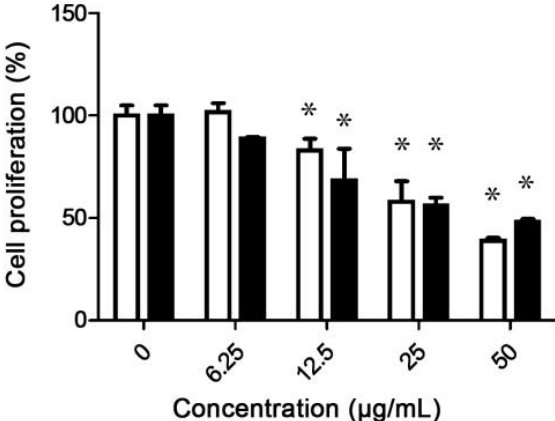

**Figure 2.** Antiproliferative activity of fraction F7 (white bars) and fraction F15 (black bars) against the A549 cell line. Asterisks indicate significant differences between treatments and respective controls ($p < 0.05$).

### 3.3. Apoptosis Induction of F7

Targeting apoptosis cell death has great relevance in cancer treatment due to its uncontrolled cell proliferation. Flow cytometry allows the detection of morphological and biochemical hallmarks in cells by apoptosis. AN and PI allow classifying the cells through the measurement of the differences in the plasmatic membrane integrity and permeability, for which the cells can be classified as viable cells ($FITC^-/PI^-$), dead cells ($FITC^-/PI^+$), early apoptosis ($FITC^+/PI^-$), and late apoptosis ($FITC^+/PI^+$) [40]. F7 was able to induce cell death by apoptosis; after 24 h of treatment, total apoptosis was 5.9% in the control vs. 19.8% in the cells treated with F7 ($p < 0.05$), with the highest percentage in late apoptosis. However, the percentage of dead cells after F7 treatment was higher than the percentage of total apoptosis. These findings suggest that F7 induces an antiproliferative effect in A549 cells through apoptotic cell death and other mechanisms that disrupt cell membrane integrity (Figure 3).

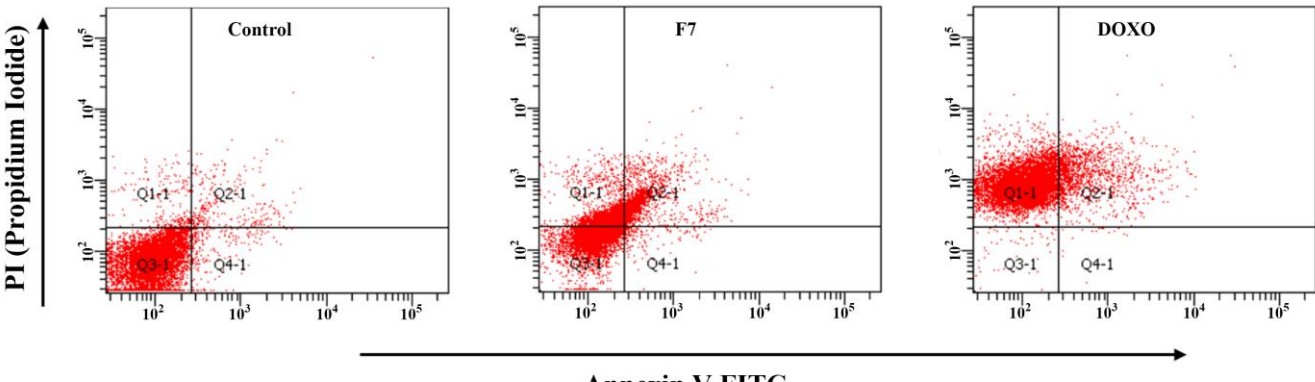

**Figure 3.** Apoptosis induction of F7 (40 μg/mL for 24 h) from *Ganoderma subincrustatum*. Dot plots represent mean ± standard deviation from three independent experiments (n = 3). Control (DMSO 0.02%), F7 (Fraction 7), DOXO (Doxorubicin hydrochloride, 1 μM).

Different ganoderic acids and extracts from *Ganoderma* spp. induce apoptosis cell death. GA T from *G. lucidum* induces mitochondria-mediated apoptosis in 95-D cancer cells (lung) with 50% of apoptotic cells at 8 h after treatment (50 μg/mL) [41]. GA H (~99% purity) induces a high apoptosis rate at 2 mg/mL after 24 h of treatment in DU-145 human prostate cancer cells [42]. An ethanol extract (25%) of *G. lucidum* shows ~30% of apoptosis at high concentration (2%) in the human gastric carcinoma AGS cell line [32]. *G. applanatum* methanol extract (80%) induces apoptosis via p53-dependent and independent pathways in Ehrlich's solid tumor [33].

The non-small cell lung cancer cell line is characterized by the overexpression of the epidermal growth factor receptor (EGFR) and its downstream effector pathways (PI3K/AKT/mTOR, RAS/RAF/MAPK, and STAT3) that promote several oncogenic processes such as cell proliferation, survival, differentiation, angiogenesis, invasion, and metastasis [26]. Xia et al. [43] reported that ganoderic acid DM induces apoptosis and autophagy in A549 and NCI-H460 through AKT/mTOR inhibition. The C-26 carboxylic group of ganoderic acid DM binds with α and β-tubulin in PC-3 (human prostate cancer) cells via an amidation reaction [44]. This suggests that the apoptotic effect of molecules in F7 could be associated with the activity modulation of these proteins by ganoderic acids in F7, particularly with disruption of the proper functioning of microtubules.

Likewise, if this molecule is produced by our strains, it can probably explain the non-selective bioactivity against different cell lines. Several bioactive molecules in certain extracts could result in high activity because components may exert bioactivity through different mechanisms of action. On the other hand, compared to the non-specific antineoplastic agent DOXO, F7 was more effective in inducing apoptosis in A549 cells ($p < 0.05$). However, the percentage of dead cells was higher in DOXO-treated cells compared to F7 ($p < 0.05$). This may be associated with the lower cytotoxicity of F7 relative to DOXO. The cytotoxicity of DOXO associated with its non-selectivity against cancer cells has been widely reported [45–47]; the clinical use of DOXO is limited by its side effects, the most dangerous is dose-dependent cumulative cardiotoxicity [48]. These findings can explain why DOXO induces greater cell death in A549 cells in relation to F7, which shows a greater apoptotic capability in relation to this reference drug. More research is needed to understand the non-selective mechanism.

Other authors have reported the ability of ganoderic acid-R (GA-R) to restore the sensitivity of multidrug-resistant (MDR) cells to DOXO via apoptosis induction. GA-R induces apoptosis in KB-A-1/Dox MDR cells in a dose-response manner from 35% to 90% after 48 h of treatment. The evaluation of the cytotoxicity showed that DOXO (0.75 μM) induces cytotoxicity less than 30% in KB-A-1/Dox cells. However, the combination of DOXO (0.75 μM) plus GA-R (5 μg/mL) significantly increased the inhibition ratio by more than 50%, which indicates that GA-R restored the sensibility of KB-A-1/Dox cells

to DOXO [49]. Although GA-R shows greater apoptotic activity in KB-A-1/Dox MDR cells compared to the apoptotic effect shown by F7 in HeLa cells, these results suggest that F7 or ganoderic acids from Gsfb can be candidates to be investigated for their possible effect against MDR cancer cells.

### 3.4. NMR Analysis of F7 Chromatographic Fraction

Lanostane-type triterpenoids are the major bioactive constituents of the *Ganoderma* genus. F7 showed characteristic displacements of lanostane-type terpenoids by NMR ($^1$H, 400 MHz; $^{13}$C, 100.6 MHz). F7 presented lanostane-type signals between 10–80, ~70, 135–150, and 195–220 ppm (Figure 4A). Lanostane-type molecules can show 0–3 signals between 195 and 220 ppm (Figure 4B). These characteristics shifts of ketone carbonyl groups can occur at C-3, C-7, C-11, C-23, and C-26 of *Ganoderma* lanostane-type compounds (Figure 4C) [13]. For each spectrum of F7, three signals above 195 ppm were observed, suggesting a maximum of three different lanostane-type compounds with a carbonyl group (Table 2). In fact, 164 ppm displacement can be explained if a lanostane is included. To know which specific molecules in F7 are exerting antiproliferative and apoptosis-inducing activity, it is necessary to design experiments for the isolation of its compounds. In addition, it is necessary to investigate the mechanisms of action of these.

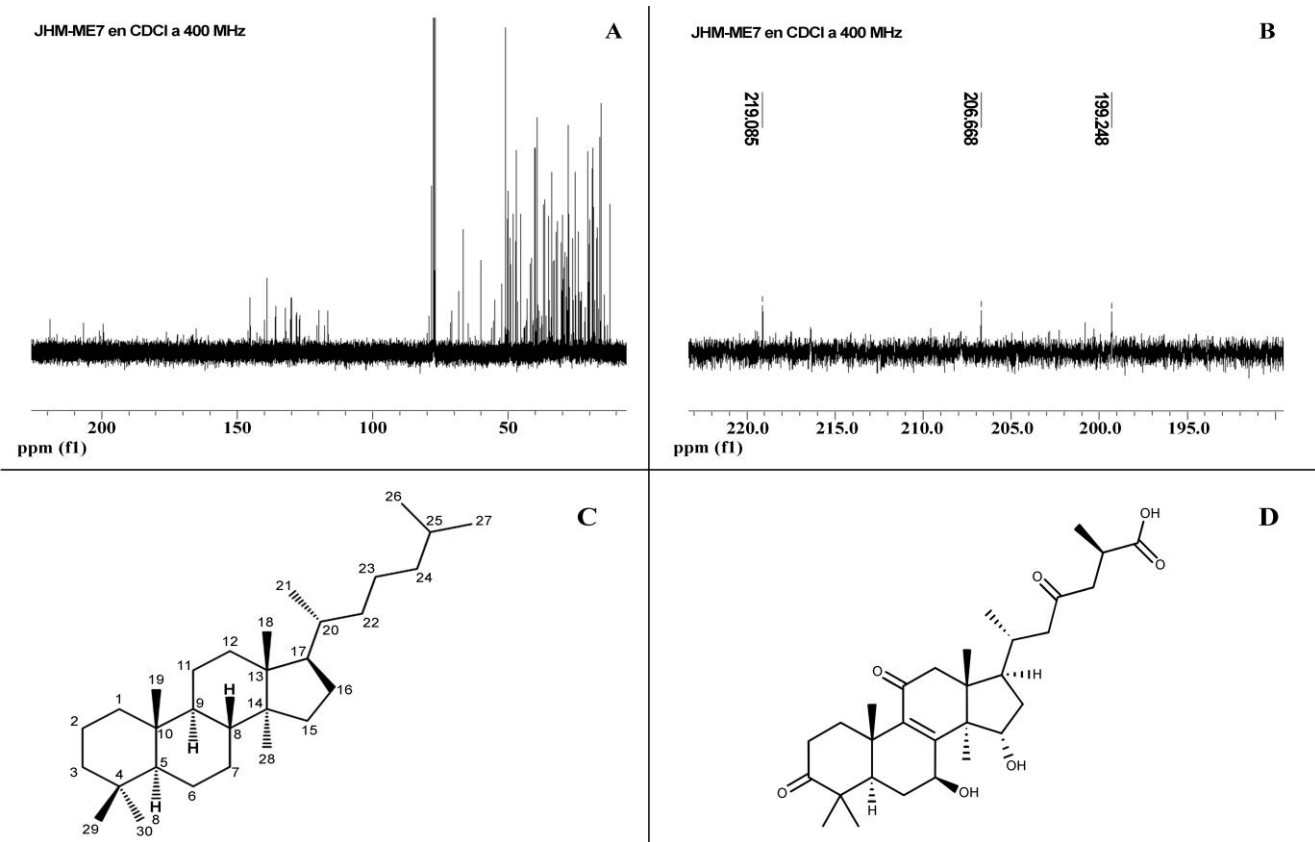

**Figure 4.** (**A**,**B**) $^{13}$C-NMR of Fraction F7 from *Ganoderma subincrustatum*. (**C**) Lanostane with α and β-unsaturated carbonyl groups. (**D**) Ganoderic acid A, one of the most abundant triterpenoids in *G. lucidum*.

Most of the *Ganoderma* highly oxygenated triterpenoids contain 27 or 30 carbon atoms, and only a few have 24. Acid triterpenoids with 27–30 carbon atoms in *Ganoderma* spp. have been isolated from mycelia, fruiting bodies, and spores, showing a carboxyl group at C26 [50]. Some of the GA isolated from fruiting bodies were A, B$_8$, C$_1$, H, K, AM$_1$, and J (Figure 4D) from spores γ, δ, ε, ζ, η, θ, and β. Moreover, GA Mi was obtained from the

mycelium of *G. lucidum*. The substituents can be at the C-3, 7, 11, 12, 15, 22, 23, 24, and 25 positions of the lanostane skeleton [13].

**Table 2.** Chemical displacement of fraction F7 from *Ganoderma subincrustatum* with a lanostane-type skeleton.

| $^{13}C$ [a] | Functional Group | C Position |
|---|---|---|
| 164.0 | C=C [b] | C-9 |
| 171.8 | COOH | C-26 |
| 199.3 | C=O | C-11 |
| 206.6 | C=O | C-23 |
| 219.1 | C=O | C-3 |

[a] Chemical shift expressed in δ values (ppm). Tetramethylsilane was used as an internal reference. Spectra recorded in CDCl$_3$. [b] Chemical displacement of C=C from C-9 in the presence of a C=O at C-7.

## 4. Conclusions

Chloroform extracts from the fruiting body of *Ganoderma subincrustatum* and *G. weberianum* were more effective in inhibiting cell proliferation compared to mycelium extracts, suggesting that the compound's profile and its concentration vary at different development stages of *Ganoderma* species. The F7 chromatographic fraction showed the highest antiproliferative activity against the A549 lung cancer cell line and had the ability to induce apoptotic cell death. NMR analysis suggests that F7 is composed of ganoderic acids due to the 171.8 signal ($^{13}C^{a}$) corresponding to a carboxyl group at C26; this could be involved in the antiproliferative and apoptotic effect of this fraction. The greater antiproliferative effect of the chloroform extracts compared to the chromatographic fractions suggests the synergistic or additive effect of the bioactive metabolites in the extracts. The strain of *G. subincrustatum* studied in this work should be used for further experiments to characterize the metabolite profile and its bioactivities and to fully understand its mechanisms of action.

**Author Contributions:** Conceptualization, D.L.-P. and M.E.; methodology, D.L.-P., H.T.-M. and M.V.-G.; validation, D.L.-P. and M.E.; formal analysis, D.L.-P., H.T.-M. and M.V.-G.; investigation, D.L.-P., A.G. and M.E.; resources, R.E.R.-Z. and M.E.; data curation, D.L.-P.; writing—original draft preparation, D.L.-P.; writing—review and editing, D.L.-P., H.T.-M., M.V.-G. and M.E.; visualization, D.L.-P., R.E.R.-Z. and M.E.; supervision, M.E. and A.G.; project administration, M.E.; funding acquisition, R.E.R.-Z. and M.E. All authors have read and agreed to the published version of the manuscript.

**Funding:** This research was funded by the National Council for Humanities, Science and Technology México (CONAHCyT A1-S-34237).

**Institutional Review Board Statement:** Not applicable.

**Informed Consent Statement:** Not applicable.

**Data Availability Statement:** Not applicable.

**Acknowledgments:** The authors are grateful to Georgina Vargas for her technical assistance in the laboratory analysis.

**Conflicts of Interest:** The authors declare no conflict of interest.

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
