# Peer review of "Antiproliferative Activity of Mycelium vs. Fruiting Body: Ganoderma subincrustatum and G. weberianum from Sonora, Mexico"

_2036-7481, doi:10.3390/microbiolres14040105_

Round 1

Reviewer 1 Report

This manuscript discusses the antiproliferative activity of chloroform extracted from mycelium or fruiting bodies of two Ganoderma species. The topic of the manuscript is interesting and appropriate for the readership of the journal. However, the manuscript should be improved in the methodology section. Additional information should be provided in subsections so that the methods can be understood and followed by the reader.

I have the following comments for improvement of the manuscript:

1.     In section 2, divide this section into two clear subsections of “2.1 Materials” and “2.2 Methods” and explain in the materials section the specifications of Ganoderma leaves (second paragraph) and list all the chemicals used for the analysis during the performed experiments. Explain thereafter the experimental procedures in the Methods section.

2.    In section 2.2 Methods, divide the explained experimental techniques and procedures into different subheadings, 2.2.1, 2.2.2, 2.2.3, etc. and explain the procedure for THC tests, NMR analysis, and column chromatography in different sections separately.

Provide details on what was measured with each technique, specifications of the devices used, details of the followed procedure and adequate referencing for the understanding of the reader. E.g. how the measurements and analysis were performed using column chromatography?

3.     Some abbreviated terms should be explained in full for  understanding of the reader:

·      In line 55, provide a complete term for TLC

·      In line 96, EtOAc-MeOH

·      In line 116, MTT

·      In  line 119, DMSO and DMEM

4.     In line 85, mention also the power used during sonication.

5.     In line 44, please check “Swallah” and if the reference is accurately cited.

6.     In lines 190-195, the antiproliferative activity of ganoderic acid is explained citing some relevant works, also the molecular structure of lanostane is shown in Figure 4. But no background information is given on lanostenoids and ganoderic acid as triterpenes beforehand. Some additional explanation should be provided in between discussions in lines 190-195 on these compounds as triterpenoids for more clarity to the reader.

Author Response

Comments and Suggestions for Authors

This manuscript discusses the antiproliferative activity of chloroform extracted from mycelium or fruiting bodies of two Ganoderma species. The topic of the manuscript is interesting and appropriate for the readership of the journal. However, the manuscript should be improved in the methodology section. Additional information should be provided in subsections so that the methods can be understood and followed by the reader.

I have the following comments for improvement of the manuscript:

  1. In section 2, divide this section into two clear subsections of “2.1 Materials” and “2.2 Methods” and explain in the materials section the specifications of Ganoderma leaves (second paragraph) and list all the chemicals used for the analysis during the performed experiments. Explain thereafter the experimental procedures in the Methods section.

Answer: Microbiology Research journal allows you to divide or combine: Materials and Methods; we prefer a section, so we kept that format. Experimental procedures were explained in more detail.

  1. In section 2.2 Methods, divide the explained experimental techniques and procedures into different subheadings, 2.2.1, 2.2.2, 2.2.3, etc. and explain the procedure for THC tests, NMR analysis, and column chromatography in different sections separately. Provide details on what was measured with each technique, specifications of the devices used, details of the followed procedure and adequate referencing for the understanding of the reader. E.g. how the measurements and analysis were performed using column chromatography.

Answer: More details about TLC test, NMR analysis, and column chromatography were indicated. 

  1. Some abbreviated terms should be explained in full for understanding of the reader:
      • In line 55, provide a complete term for TLC

Answer: Line 121, it was indicated

      • In line 96, EtOAc-MeOH

Answer: Line 119, it was indicated

      • In line 116, MTT

Answer: Line 147, it was indicated

      • In line 119, DMSO and DMEM

Answer: Line 152, DMSO was indicated; Line 136, DMEM was indicated

  1. In line 85, mention also the power used during sonication.

Answer: Line 108, power was indicated

  1. In line 44, please check “Swallah” and if the reference is accurately cited.

Answer: Line 65, it was corrected

  1. In lines 190-195, the antiproliferative activity of ganoderic acid is explained citing some relevant works, also the molecular structure of lanostane is shown in Figure 4. But no background information is given on lanostenoids and ganoderic acid as triterpenes beforehand. Some additional explanation should be provided in between discussions in lines 190-195 on these compounds as triterpenoids for more clarity to the reader.

Answers: In Lines 54-67 and Lines 274-281, the required information was included

Reviewer 2 Report

The authors have studied the Antiproliferative Activity of chloroform extract of Mycelium and Fruiting Body from Ganoderma subincrustatum and G. weberianum, including Apoptosis studies. The current version of the manuscript is suitable for publication; however, the authors need to address the following comments.

1.

Did the authors have to isolate any novel compounds from chloroform extracts from G. subincrustatum and G. weberianum? It would be good if the authors included the list of compounds, as it would strengthen the manuscript.

2.

What are the major chemical constituents of chloroform extract, in addition to ganoderic acid, to show a promising effect of antiproliferative activity?  

3.

Introduction: The authors should write the biological importance of triterpenoids in drug discovery. Please refer to the following articles. EJMC 188, 2020, 111974 (https://doi.org/10.1016/j.ejmech.2019.111974) & EJMC114, 2016, 293-307 (https://doi.org/10.1016/j.ejmech.2016.03.013)

4.

Please include a structure of Ganoderic acid. And also, compare the NMR data of the new product and Ganoderic acid using the table.

5.

Provide the HRMS data for F7 compound.

NA

Author Response

 Comments and Suggestions for Authors

The authors have studied the Antiproliferative Activity of chloroform extract of Mycelium and Fruiting Body from Ganoderma subincrustatum and G. weberianum, including Apoptosis studies. The current version of the manuscript is suitable for publication; however, the authors need to address the following comments.

  1. Did the authors have to isolate any novel compounds from chloroform extracts from G. subincrustatum and G. weberianum? It would be good if the authors included the list of compounds, as it would strengthen the manuscript.

Answer: F7 showed characteristic displacements of lanostane-type terpenoids by NMR, but compounds are still undetermined (Lines 305-310)

  1. What are the major chemical constituents of chloroform extract, in addition to ganoderic acid, to show a promising effect of antiproliferative activity?  

Answer: Most of the Ganoderma highly oxygenated triterpenoids contain 27 or 30 carbon atoms, only a few have 24. Acid triterpenoids with 27-30 carbon atoms in Ganoderma spp. have been isolated from mycelia, fruiting bodies, and spores, showing a carboxyl group at C26 (Lines 317-323). All of them have potential antiproliferative activity.

  1. Introduction: The authors should write the biological importance of triterpenoids in drug discovery. Please refer to the following articles. EJMC 188, 2020, 111974 (https://doi.org/10.1016/j.ejmech.2019.111974) & EJMC114, 2016, 293-307 (https://doi.org/10.1016/j.ejmech.2016.03.013).

Answer: Both references were included (Lines 241-247; Lines 456-461). Likewise, articles on triterpenoids from Ganoderma and a couple of plant species, as well as a couple of nanoemulsions with antiproliferative activity vs. A549 cell line were included (Lines 429-465)

  1. Please include a structure of Ganoderic acid. And also, compare the NMR data of the new product and Ganoderic acid using the table.

Answer: NMR analysis suggested that F7 fraction is composed of lanostane-type terpenoids, but ganoderic acids were not determined; so, lanostane and ganoderic acid A structure were included as an example (Figs. 4C, 4D) and its variability explained in the text (Lines 309-310, 317-323)

  1. Provide the HRMS data for F7 compound.

Answers: F7 compounds were not determined

Round 2

Reviewer 1 Report

This manuscript discusses and compares the antiproliferative activity of triterpenoids extracted from mycelium and fruiting bodies of two Ganoderma species. Column fractionation was performed for the species showing higher antiproliferative activity to study the apoptosis induction of each fraction. The study was followed by NMR analysis for the fractions showing higher apoptosis induction to obtain their composition and the bioactive triterpenoid compound contributing the most to apoptotic cell death.

The topic of the work is appropriate for the readership of the journal and the manuscript reads well. However, the manuscript must be improved considering the following points:

1.     In line 108, the authors provided the specification of ultrasonication, “power of 40 KHz”. "power" should change to "frequency". Please also provide the power used for ultrasonication in the unit watts (W).

2.    In line 303, The authors show in Figure 2 that F7 fraction showed a higher antiproliferative activity compared with the control (Doxorubicin) but they state that the number of dead cells was higher for Doxo-treated cells in comparison. The authors explain it by the probable lower cytotoxicity of F7 compared with DOXO.

Please see the relevant work below:

https://doi.org/10.4028/www.scientific.net/AMR.834-836.573

In this work (see Figure 2) Ouyang et al. study and compare the antiproliferative activity and cytotoxicity effect of ganoderic acid when used together with doxorubicin on tumour cells or multidrug-resistant tumour cells.

Please comment and add an additional explanation in the manuscript on how your findings compare with theirs.

Author Response

Thank you very much for reviewing this manuscript. Please find the detailed responses below and the corresponding revisions/corrections highlighted in yellow in the second corrected version.

Comment 1: In line 108, the authors provided the specification of ultrasonication, “power of 40 KHz”. "power" should change to "frequency". Please also provide the power used for ultrasonication in the unit watts (W).

Response: Both were indicated: Lines 107-108: … with a power of 210.6 W and operating frequency of 40 kHz.

Comments 2: In line 303, The authors show in Figure 2 that F7 fraction showed a higher antiproliferative activity compared with the control (Doxorubicin) but they state that the number of dead cells was higher for Doxo-treated cells in comparison. The authors explain it by the probable lower cytotoxicity of F7 compared with DOXO.

Response: It was commented on why DOXO could be higher cytotoxic: Lines 301-304: The cytotoxicity of DOXO associated with its non-selectivity against cancer cells has been widely reported [46,47,48]; the clinical use of DOXO is limited by its side effects, the most dangerous is dose-dependent cumulative cardiotoxicity [49].

Comments 3: Please see the relevant work below: https://doi.org/10.4028/www.scientific.net/AMR.834-836.573 In this work (see Figure 2) Ouyang et al. study and compare the antiproliferative activity and cytotoxicity effect of ganoderic acid when used together with doxorubicin on tumour cells or multidrug-resistant tumour cells. Please comment and add an additional explanation in the manuscript on how your findings compare with theirs.

Response: Article: Ganoderic acid restores the sensitivity of multidrug resistance cancer cells to doxorubicin by Ouyang et al. (2014) was consulted. The most important results of Figure 2 are presented and compared with F7. Lines 308-318: Other authors have reported the ability of ganoderic acid-R (GA-R) to restore the sensitivity of multidrug-resistant (MDR) cells to DOXO via apoptosis induction. GA-R induces apoptosis in KB-A-1/Dox MDR cells in a dose-response manner from 35 to 90% after 48 h of treatment. The evaluation of the cytotoxicity showed that DOXO (0.75 µM) induces cytotoxicity less than 30% in KB-A-1/Dox cells. However, the combination of DOXO (0.75 µM) plus GA-R (5 µg/mL) significantly increased the inhibition ratio by more than 50%, which indicates that GA-R restored the sensibility of KB-A-1/Dox cells to DOXO [50]. Although GA-R shows greater apoptotic activity in KB-A-1/Dox MDR cells compared to the apoptotic effect shown by F7 in HeLa cells, these results suggest that F7 or ganoderic acids from Gsfb could be candidates to be investigated for their possible effect against MDR cancer cells.

Five new references were included: Lines 496-507:

[46] Zhang, Q.; Wu., L. In vitro and in vivo cardioprotective effects of curcumin against doxorubicin-induced cardiotoxicity: a systematic review. J. Oncol. 2022, 2022:7277562. https://doi.org/10.1155/2022/7277562.

[47] Supasena, W.; Muangnoi, C.; Praengam, K.; Wui Wong, T.; Qiu, G.; Ye, S.; Wu, J.; Tanasupawat, S.; Rojsitthisak, P. Enhanced selective cytotoxicity of doxorubicin to breast cancer cells by methoxypolyethylene glycol conjugation via a novel beta-thiopropanamide linker. Eur. Polym. J. 2020, 141:110056. https://doi.org/10.1016/j.eurpolymj.2020.110056.

[48] Osman, A.M.M.; Bayoumi, H.M.; Al-Harthi, S.E.; Damanhouri, Z.A.; ElShal, M.F. Modulation of doxorubicin cytotoxicity by resveratrol in a human breast cancer cell line. Cancer Cell. Int. 2012, 12:47. https://doi.org/10.1186/1475-2867-12-47.

[49] Torres-Moreno, H.; Marcotullio, M.C.; Velazquez, C.; Arenas-Luna, V.M.; Hernández-Gutiérrez, S.; Robles-Zepeda, R.E. Cucurbitacin IIb from Ibervillea sonorae induces apoptosis and cell cycle arrest via STAT3 inhibition. Anticancer Agents Med. Chem. 2020, 20(10):1188-1196. https://doi.org/10.2174/1871520620666200415101701.

[50] Ouyang, J.J.; Wang, Y.Q.; Tang, W. Ganoderic acid restores the sensitivity of multidrug resistance cancer cells to doxorubicin. Adv. Mater. Res. 2014, 834:573–576. https://doi.org/10.4028/www.scientific.net/AMR.834-836.573.